# PROCEEDINGS A

mathematical modelling

hand hygiene, respiratory infections, community, influenza, modelling

**Author for correspondence:**
Thi Mui Pham
e-mail: thi.mui.pham@posteo.de

# The potential impact of intensified community hand hygiene interventions on respiratory tract infections: a modelling study

Thi Mui Pham[1], Mo Yin[2,3,4] and Ben S. Cooper[2,3]

[1]Julius Center for Health Sciences and Primary Care, University Medical Center Utrecht, Utrecht University, Utrecht, The Netherlands
[2]Oxford Centre for Global Health Research, Nuffield Department of Medicine, University of Oxford, Oxford, United Kingdom
[3]Mahidol-Oxford Tropical Medicine Research Unit, Mahidol University, Bangkok, Thailand
[4]Division of Infectious Diseases, University Medicine Cluster, National University Hospital of Singapore, Singapore

TMP, 0000-0001-6748-2479

Hand hygiene is among the most fundamental and widely used behavioural measures to reduce the person-to-person spread of human pathogens and its effectiveness as a community intervention is supported by evidence from randomized trials. However, a theoretical understanding of the relationship between hand hygiene frequency and change in risk of infection is lacking. Using a simple model-based framework for understanding the determinants of hand hygiene effectiveness in preventing viral respiratory tract infections, we show that a crucial, but overlooked, determinant of the relationship between hand hygiene frequency and risk of infection via indirect transmission is persistence of viable virus on hands. If persistence is short, as has been reported for influenza, hand-washing needs to be performed very frequently or immediately after hand contamination to substantially reduce the probability of infection. When viable virus survival is longer (e.g. in the presence of mucus or for some enveloped

viruses) less frequent hand washing can substantially reduce the infection probability. Immediate hand washing after contamination is consistently more effective than at fixed-time intervals. Our study highlights that recommendations on hand hygiene should be tailored to persistence of viable virus on hands and that more detailed empirical investigations are needed to help optimize this key intervention.

## 1. Introduction

Promotion of hand hygiene is a key public health intervention in preventing the spread of infectious diseases. Since the mid-1800s, when Ignaz Philip Semmelweis demonstrated that hand washing could dramatically reduce maternal mortality due to puerperal fever [1], hand hygiene has been the cornerstone of infection prevention and control policies. In hospital settings, hand hygiene has played a major role in successfully controlling hospital-acquired infections, especially those caused by methicillin-resistant *Staphylococcus aureus* [2]. In the community, there is evidence from randomized controlled trials that hand hygiene interventions can be effective in reducing both the risk of diarrhoeal disease [3] and respiratory tract infections [4–6].

Hand hygiene is simple, low-cost, minimally disruptive and, when widely adopted, may lead to substantial population-level effects [5,7]. While randomized controlled trials of hand hygiene interventions in the community provide evidence that such interventions are effective in reducing the incidence of respiratory tract infections, reported effect sizes are highly variable [4,6]. It is unclear to what extent this variability is explained by success in achieving substantial changes in hand hygiene behaviour in these trials. Understanding how the effectiveness of hand hygiene in reducing transmission scales with hand hygiene frequency is important for assessing the extent to which interventions that aim at achieving a large and sustained increase in community hand hygiene can contribute to infection suppression.

Fomite-mediated transmission has been modelled for various respiratory viruses such as SARS-CoV-2 [8,9] and influenza [10–14]. These studies usually involve a Quantitative Microbial Risk Assessment (QMRA) approach that explicitly models the level of contamination on hands and surfaces assuming a dose–response relationship. This approach leads to realistically detailed models that are particularly useful if detailed environmental data and respective parameters are available. Several studies have also evaluated the effect of hand hygiene on the transmission risk and compared hand hygiene interventions to surface disinfection (e.g. [9,12]). While hand hygiene compliance was taken into account, none of these previous studies compared different timings and hand hygiene strategies.

In this study, we took a theory-based approach and developed a simple mechanistic mathematical model to understand the relationships between the various components of respiratory tract infection transmission pathways involving hand contamination. We aimed to quantify the expected impact of different hand hygiene behaviours on risks of respiratory tract infection. Our work is motivated by published data on the survival of influenza A on human fingers. We therefore focus on viral respiratory tract infections but our model also applies to pathogens for which similar assumptions apply. Finally, we consider the implications of the outcomes of these analyses for the potential contribution of intensifying community hand hygiene to the suppression of respiratory tract infections.

## 2. Methods

### (a) Overview

We consider human pathogens where transmission is mediated by contaminated hands. We neglect direct droplet and aerosol transmission. Hands are assumed to become contaminated with infectious material via contact with contaminated surfaces or an infected person. In the

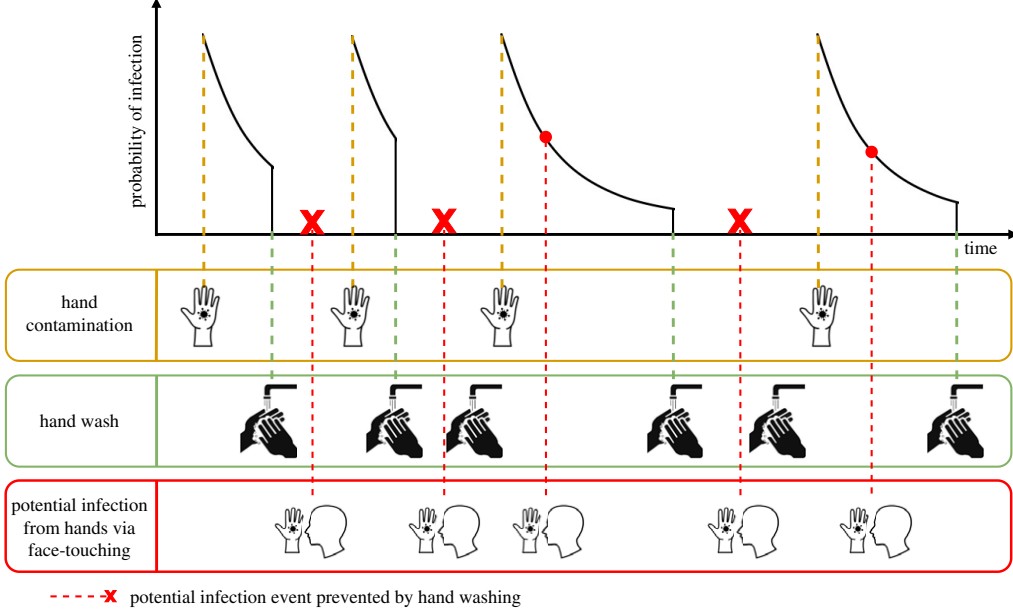

**Figure 1.** Hand hygiene model. Illustration of potential infection events from hands via face-touching, hand contamination events and hand washing events. Hand contamination events cause a stepwise increase in the probability of infection resulting from face touching events, which then decreases exponentially with time. Effective hand washing reduces the probability of infection to zero during subsequent face touching if no further hand contamination events occur. An infection may occur between a hand contamination event and hand washing, depending on the probability of infection at the moment of face touching. (Online version in colour.)

absence of hand washing, hands do not remain contaminated indefinitely; instead, as has been shown experimentally, the probability of remaining contaminated and capable of transmitting infection declines over time (figure 1, top panel) [15,16]. If contaminated hands of a susceptible host make contact with the host's mucous membranes in the eyes, nose or mouth there is some probability of the host becoming infected. Effective hand washing interrupts this process by removing viable virus from the hands. An immediate consequence of this conceptualization is that the time interval between the hands becoming contaminated and the potential transmission to the host can have a critical impact on how effective a given frequency of hand washing will be at interrupting transmission (figure 2). Given a certain probability of infection, the time interval between hand contamination and transmission to the host's mucosa tends to be longer if pathogen persistence on hands is long and vice-versa. If this time interval is relatively long, i.e. the virus survives on hands for a long time, regular effective hand hygiene will have a high chance of blocking potential transmission events (red crosses in figure 2*a*) in the absence of hand hygiene. By contrast, if this time interval is short, i.e. the pathogen persists for only a short amount of time, much more frequent hand hygiene will be needed to block an appreciable proportion of transmission events (figure 2*b*).

## (b) Hand hygiene scenarios

We explored the effect of hand hygiene on the probability of infection and considered two hand washing schemes that are distinguished by different timings of hand washing:

 (i) fixed-time hand washing (uniformly at fixed time intervals),
 (ii) event-prompted hand washing (with a delay after hand contamination events).

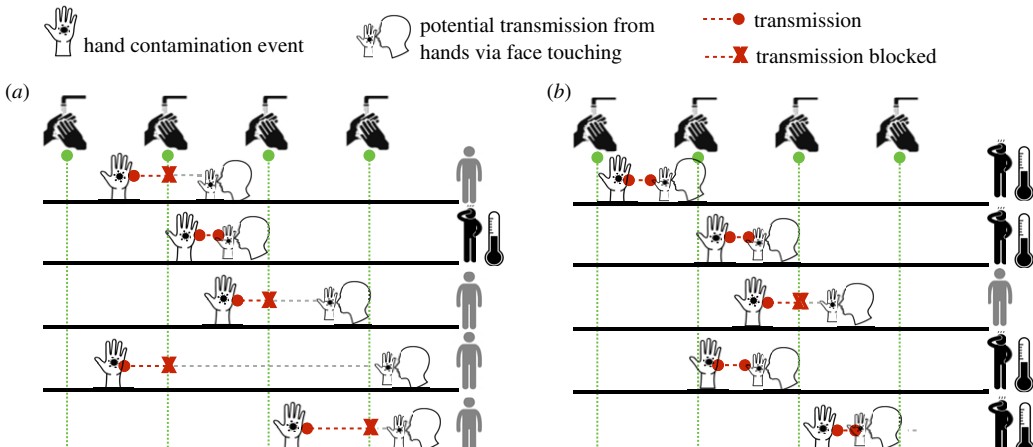

**Figure 2.** Long versus short time interval between hand contamination and infection with regular hand washing. (*a*) When there are long time intervals between hand contamination and potential infection from hands via face touching, hand washing can block many infection events and substantially reduce the risk of infection. (*b*) When there are short time intervals between hand contamination and face touching, it is likely that hand washing can disrupt only a few infections. (Online version in colour.)

## (c) Mathematical model

We assumed that hands of susceptible individuals become contaminated at random. These contamination events are assumed to occur independently of each other, and to follow a Poisson distribution with a mean of $\lambda_c$ events per hour. Once contaminated, we assumed that in the absence of hand washing there is a constant rate at which hands get decontaminated. Thus, the probability of the virus persisting on hands at time $t$ after contamination, $P(t)$, is assumed to decay exponentially with a half-life of $T_{1/2}$. This is consistent with experimental data for influenza A [16]. We further assumed that, in a given time interval $[0, T]$, individuals touch their face at random times $t_1, \ldots, t_F$ leading to potential self-infection events that are assumed to occur independently of each other, and follow a Poisson distribution with a mean of $\lambda_f$ events per hour. The probability that a single face touching contact with contaminated hands actually leads to transmission is denoted by $\epsilon$. This parameter represents the transfer efficiency from fomites to hands and combines several factors such as the level of contamination on hands, the infectivity of the virus and the susceptibility of the host. We do not explicitly model a dose–response relationship, thus $\epsilon$ represents the average probability of infection per hand-mucosa contact. In the case of a sequence of hand contamination events, we assumed that the probability $P(t)$ is reset to its baseline value of one and that the transmission probability $\epsilon$ is kept the same. The probability $P(t)$ is assumed to always have the same functional shape even after sequential hand contamination events. We, therefore, do not account for accumulation of infection risks for heterogeneity of hand contamination events. The force of infection that a susceptible individual at time $t$ becomes infected is, therefore, $\lambda_{\text{inf}}(t) = \epsilon P(t)$. The cumulative probability of infection over a given time period $T$ is then given by: $1 - e^{-\sum_{i=1}^{F} \epsilon P(t_i)}$. We assumed that when hand washing is performed after the last hand contamination event and before a face touching event at time $t_i$, the respective probability of pathogen persistence $P(t_i)$ is reduced to zero. Thus, hand washing is assumed to remove all virus on contaminated hands completely after one wash, regardless of the number of hand contamination events that took place between hand washing events. A more detailed mathematical description of the model is included in the electronic supplementary material, p. 15.

## (d) Parameters

When available, parameter estimates were obtained from the literature. Otherwise, we performed sensitivity analyses where parameters were varied within plausible ranges (table 1).

The probability of transmission per face touching event, $\epsilon$, was constrained to meet a fixed probability of infection to reflect the fact that we are interested in how our beliefs about the potential impact of enhanced hand hygiene for a pathogen of known transmissibility will vary according to what we know or believe about its survival on hands. In our main analysis, we assumed a cumulative probability of infection of 10% over a time period of 12 h. This is roughly based on secondary attack rates for influenza, influenza-like illnesses and acute respiratory illness in household studies [23–26] accounting for the relative contribution of fomite-mediated transmission [10,14,27]. Note that high uncertainty lies around these parameters as the quantification of routes of transmission remains difficult [13,28,29]. In sensitivity analyses, we examine the results for cumulative probabilities of infection of 0.1%, 1%, 5%, 30% and 50%. By covering such a broad range, we ensure that our conclusions are robust to this parameter that highly depends on the setting and the pathogen.

In the fixed-time hand washing scheme, we varied time intervals between hand washing to be 5 min to 6 h. For event-prompted hand washing, the delay of hand washing after hand contamination events was varied from 1 min to 6 h.

There is little published data on the rate of hand contamination events susceptible individuals are exposed to when in contact with infected individuals who are shedding respiratory viruses. In a direct observation study conducted by Zhang *et al.* [21], surface touching behaviour in a graduate student office was recorded. Approximately 112 surface touches per hour were registered. Another study by Boone *et al.* [22] found that the influenza virus was detected on 53% of commonly touched surfaces in homes with infected children (using reverse transcriptase-polymerase chain reaction (RT-PCR)). Informed by these values, we used 60 events per hour as the upper bound for the rate of hand contamination events $\lambda_c$. Note that $\lambda_c = 60\,\mathrm{h}^{-1}$ is based on a RNA to viable virus ratio of 1:1 and should be seen as a theoretical upper bound as in practice, this ratio is likely much smaller. We chose one hand contamination event per hour as the lower bound. In our main analyses, we used a rate of four hand contamination events per hour.

In [16], the survival of influenza A on human fingers was experimentally investigated. We fitted exponential decay curves to these results in order to determine the half-life of the probability of persistence of H3N2 for two viral volumes, 2 µL and 30 µl (table 1 and electronic supplementary material). We use these values as examples for the half-life of the probability of pathogen persistence. In addition, we vary the half-life of the probability of persistence from 1 to 60 min in our analysis.

## (e) Model analyses and outcomes

The model output is the cumulative probability of a susceptible person becoming infected in 12 hours and we will refer to it subsequently as simply the probability of infection. We investigated the impact of hand washing on the probability of infection for different hand contamination rates. In addition, we compared the two hand washing schemes (fixed-time vs. event-prompted) to find the optimal hand washing strategy that will lead to the greatest reduction of the probability of infection. The model was implemented in R v. 3.6.3 [30]. The code reproducing the results of this study is available at https://github.com/tm-pham/handhygiene_modelling [31].

# 3. Results

## (a) Impact of half-life of pathogen persistence on probability of infection

Viral persistence on hands plays a key role on the effect of increasing hand hygiene frequency. The longer the virus survives on the hands, the larger the impact of increasing hand washing uptake on the probability of infection. For example, when the half-life of viral persistence is 1 min,

**Table 1.** Parameter values.

| | | value | source |
|---|---|---|---|
| time period | | 12 h | assumed |
| rate of infection events through face touching (per hour) | $\lambda_f$ | 10 (5, 20, 50) | [17–20][a] |
| cumulative probability of infection (in 12 h) | | 10% (0.1%, 1%, 5%, 30%, 50%)[b] | assumed |
| probability of transmission per face touching event | $\epsilon$ | computed from cumulative probability of infection | |
| rate of hand contamination events (per hour) | $\lambda_c$ | 4 h$^{-1}$ (1, 20, 60 h$^{-1}$)[b] | [21,22] |
| time between hand washing events (fixed-time) | $t_F$ | 5, 15, 30 min, 1 h, 2, 6 h | assumed |
| delay of hand washing after hand contamination events | $t_D$ | 1, 5, 15, 45 min, 1 h, 2, 6 h | assumed |
| half-life of virus persistence | $T_{1/2}$ | 1–60 min | varied |
| half-life of H3N2 persistence for 2 μl of viral inoculum | | 5.4 min | [16] |
| half-life of H3N2 persistence for 30 μl of viral inoculum | | 36.1 min | [16] |

[a]Mean face touching frequency involving mucous membranes (eyes, mouth, nose).
[b]Sensitivity analyses.

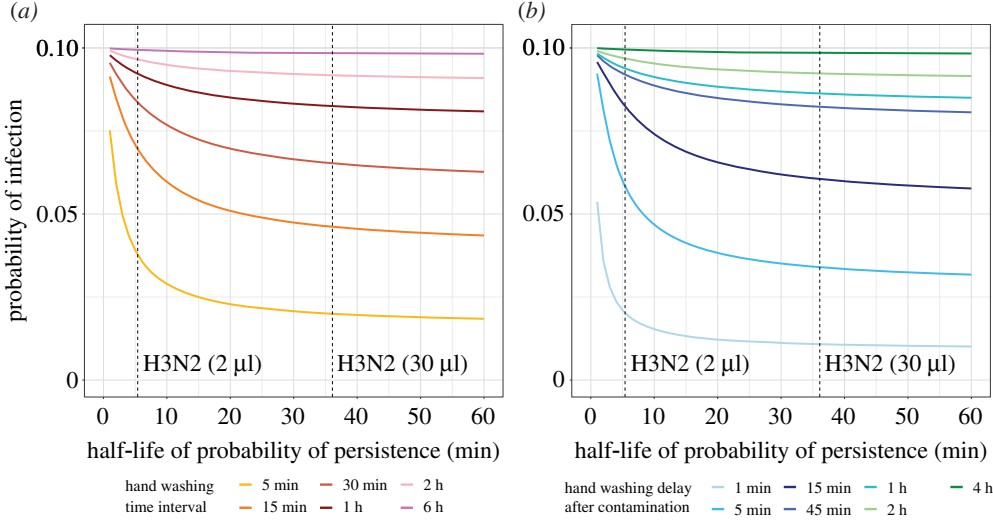

**Figure 3.** Impact of half-life of viral persistence on probability of infection for different hand washing schemes and frequencies. (*a*) Fixed-time hand washing (*b*) event-prompted hand washing. In this graph, we assumed that a susceptible individual is exposed to a baseline probability of infection of 10% if no hand washing is performed within the time period of 12 hours. The dashed lines represent the half-life of viral persistence for H3N2 inoculum volumes of 2 μl and 30 μl (calculated from [16]). For each half-life value, the probability of transmission per face touching event $\epsilon$ was determined for a probability of infection of 10% in the case of no hand washing. The probability of infection for the different hand washing frequencies/delays was then computed using this $\epsilon$ value. Hand contamination events are assumed to occur on average four times per hour. Sensitivity analyses with different values for baseline probabilities of infection as well as the half-life calculations are presented in the electronic supplementary material. (Online version in colour.)

hand washing every 15 min reduces the probability of infection from 10% to 9.2% (figure 3*a*). When the half-lives increase to 5.4 min and 36.1 min (equivalent to the half-lives of H3N2 persistence of 2 μl and 30 μl viral inoculum, respectively), the same hand washing frequency decreases the probability of infection to 6.9% and to 4.6%, respectively. Consequently, fewer hand washes are necessary to reduce the probability of infection by 50% for long compared with short durations of viral persistence (see figure S2). This observation can be explained by the fact that the shorter the virus persists on hands, the shorter the intervals between hand contamination and transmission events tend to be (with a higher transmission probability per contact needed for the same cumulative probability of infection, see figure S3) and, therefore, the less likely hand washing is able to interrupt infection events. Figure S2 shows that the delay between hand contamination and hand washing needed to prevent 50% of transmissions is shorter when the half-life of viral persistence on the hands is shorter, confirming the hypothesis that timely hand washing is especially crucial if the virus survives only a short time on hands. Furthermore, the effect of hand washing on reducing the probability of infection plateaus with increasing duration of virus persistence (figure 3). This can be attributed to the hand contamination rate, i.e. new events occur before the virus decays.

## (b) Comparison of hand washing schemes

The second notable finding from the model is that event-prompted hand washing is more effective than fixed-time hand washing in reducing the probability of infection. We illustrate this in figure 4 by comparing both schemes using four different hand washing frequencies/delays, each with approximately the same average number of hand washing events performed per hour. For example, hand washing regularly every 15 min is compared to event-prompted hand washing

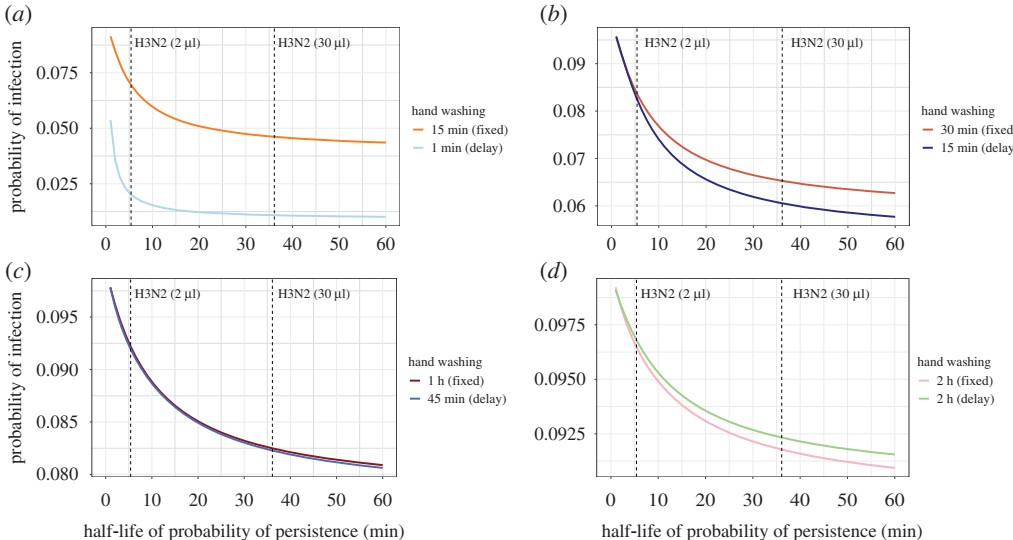

**Figure 4.** Comparison of the impact of the two hand washing schemes on the cumulative probability of infection. Hand washing at fixed time intervals and event-prompted hand washing (with a time delay) with similar average number of hand washing events per hour are compared for a hand contamination rate of $\lambda_c = 4\,\mathrm{h}^{-1}$. A baseline probability of infection of 10% is assumed when there is no hand washing. The dashed lines represent the half-life values of H3N2 persistence for 2 µl and 30 µl inoculum volumes [16]. (Online version in colour.)

1 min after each hand contamination event (set at four per hour). If the half-life of viral persistence is similar to 2 µl of H3N2 inoculum ($T_{1/2} = 5.4\,\mathrm{min}$), the baseline probability of infection of 10% (no hand washing) is reduced to about 6% and 2% when hand washing is performed every 15 min and one minute after hand contamination events, respectively. The differences between the two hand washing schemes are less pronounced if hand washing is performed less frequently or with a longer delay after hand contamination events since the two hand washing schemes become more similar. It follows that delays between hand contamination and hand washing decrease the effect of hand washing on reducing the probability of infection.

## (c) Hand contamination rate

Another important parameter that affects the effect of hand hygiene is the hand contamination rate. Figure 5 shows the increase in hand hygiene frequency required to halve the probability of infection from 10% (no hand washing) to 5%. When the hand contamination rate is relatively low (i.e. less than 10 contamination events per hour), fewer hand washes are needed to reduce the probability of infection if hand washing is event-prompted. In addition, the longer the virus persists on hands, the smaller the number of hand washing events necessary to achieve a given reduction in the probability of infection. This effect is less pronounced for event-prompted than for time-fixed hand washing, re-emphasizing the finding that when hand contamination events occur very frequently, hand washing would need to be very frequent to have a substantial impact on reducing the probability of infection (e.g. at least five times per hour to prevent 50% of transmission in the case of a half-life of 36.1 min). In this case, susceptible individuals are exposed to a continuous risk of hand contamination and hand washing has only a limited impact on reducing the risk of infection.

Our qualitative conclusions do not change with respect to different baseline probabilities of infection and hand contamination rates (see electronic supplementary material for sensitivity analyses with respect to these parameters).

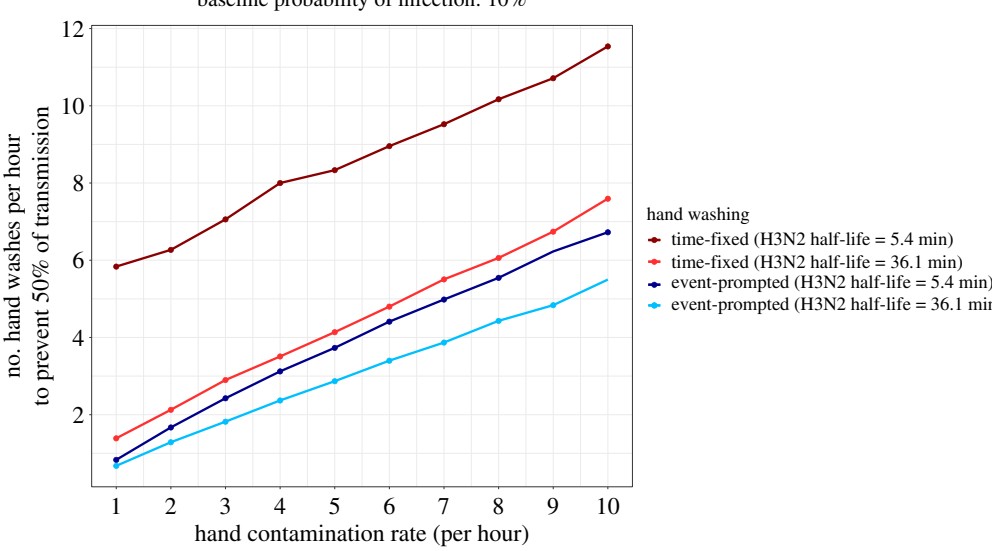

**Figure 5.** Number of hand washes necessary to prevent 50% of transmissions. For a baseline probability of infection of 10%, the number of hand washing events necessary to reduce the probability of infection to 5% was computed for time-fixed and event-prompted hand washing and a range of hand contamination rates. We used the half-life of H3N2 persistence for viral inoculum volumes of 2 μl and 30 μl (calculated from [16]). (Online version in colour.)

## 4. Discussion

Our study provides new insights into factors that affect the effectiveness of hand hygiene behaviour in reducing the probability of infection. Firstly, we found that the shorter the virus survives on hands, the less effective increasing hand washing frequency is in reducing infection. The logic behind this is that when the virus dies off quickly before hand washing is performed, the time intervals between hand contamination and transmission tend to be shorter and the respective transmission probability per contact needs to be higher for the same cumulative probability of infection. Secondly, the contact frequency with contaminated surfaces is crucial for the effect of hand washing. The more often hands become contaminated, the more frequently hands need to be washed to reduce infection risk. Lastly, when hands are not constantly contaminated, event-prompted hand washing is more efficient than fixed-time hand washing given the same hand washing frequency. This is because delays in hand washing after contamination of hands in fixed-time compared to event-prompted hand washing tend to be longer, and, during this delay, susceptible hosts may become infected through face touching.

These findings provide additional insights into the modest and heterogeneous effects of hand hygiene reported by hand hygiene trials aimed at reducing respiratory tract infections in the community [4,6,32], and also provide pointers to potentially more effective hand hygiene interventions. These trials are challenging to conduct due to the difficulties in implementing behaviour change, including poor adherence to hand washing recommendations [33], and loss to follow-up [34,35]. However, given the low cost and minimally disruptive nature of the intervention we believe there would be considerable value in building on this experimental work and the theory outlined above to develop improved hand-hygiene interventions. This could offer considerable public health benefit both in interpandemic and pandemic periods.

Since the hand contamination rate directly impacts the effect of hand hygiene, specific hand hygiene advice should cater for different situations where surface contamination differs markedly. For example, contacts in the community and in a household with an infectious person would likely result in very different hand contamination rates. In the first case, where hand

contamination events occur at a moderate rate, hand washing needs to be performed frequently or immediately after hand contamination events in order to substantially reduce the probability of infection. While individuals may not always be aware of all hand contamination events, event-prompted hand washing can be facilitated by installing or providing hand sanitizers in public areas with high-touch surface areas, such as public transportation and supermarkets, to reduce the delay in hand cleansing. Furthermore, in the second case, where hands become contaminated very frequently, a substantial reduction in the probability of infection is unlikely to be attained unless hand washing frequency is increased drastically, i.e. every one to five minutes. Because hand washing at such a high rate is not practical (neither for fixed-time nor event-prompted hand washing), the recommendations in this scenario are to regularly clean the environment (such as surfaces), reduce the rate of surface touching (if possible), and/or isolate infected individuals to reduce hand contamination events.

We performed sensitivity analyses with varying parameter values and distributions to ensure our conclusions are qualitatively robust. Nevertheless, our results have several limitations which reflect our decision to use a simple model that subsumes much of the biological complexity into a few basic parameters.

We specifically modelled indirect transmission routes via hands and did not consider direct droplet and aerosol transmission. To date, there is little known about the relative importance of the various transmission routes of respiratory pathogens [10,28,36,37]. In particular, the relative contribution of each route may depend on the pathogen itself, the setting and environmental conditions [29]. For example, in a two-route transmission model, the fomite route was estimated to contribute 6% to the infections caused by influenza in a nosocomial outbreak in Hong Kong [38]. For SARS-CoV-2, Azimi and colleagues developed a mechanistic transmission modelling framework with multiple transmission routes and used detailed information available from the Diamond Princess cruise ship outbreak [39]. The fomite-mediated mode was estimated to contribute to 21% (median) of infected cases aboard the ship. In a healthcare setting, the contribution of contact transmission was estimated to contribute 8.2% (0.0, 0.37%) to the overall infection risk for SARS-CoV-2 [40]. In a comparative analysis of outbreaks of influenza H1N1, SARS-CoV, and norovirus in aircabins, the fomite route was estimated to play the dominant role for SARS-CoV (50%, 95% CI: 48–53%) while its contribution was minor for H1N1 (less than 1%) [41]. When other routes are considered, the baseline probability of infection needs to be adapted with lower values leading to a lower absolute effect of hand hygiene.

We assumed no accumulation of the probability of infection for a sequence of hand contamination events and did not account for heterogeneity of hand contamination events. Instead, the probability of viral persistence is reset to its baseline value of one and the transmission probability $\epsilon$ is kept equal for consecutive hand contamination events. Our model, therefore, might underestimate the reduction in the probability of infection induced by hand washing. However, since we do not expect this to affect the hand washing schemes differently, we do not expect this to change our qualitative conclusions.

We did not account for variation and likely auto-correlation in the probability of infection per hand-mucosa contact or for non-homogeneity in the rate of hand contamination. While incorporating such factors would have some effect on precise quantitative results reported, they would not affect the broad qualitative conclusions. Further analyses included in our electronic supplementary material show that the relationship between the hand contamination rate and viral survival on hands is an important determinant for evaluating the effect of hand washing on the risk of fomite-mediated infection. This relationship remains underexplored and further investigations are needed for each pathogen in question to reliably estimate the potential impact of hand washing.

We assumed that hand washing reduced the probability of persistence on hands to zero and hence that hand washing is maximally efficacious in removing the virus from hands. Although a high efficacy of hand hygiene with soap and water and alcohol-based hand rubs has been demonstrated [42,43], its real world effectiveness depends on how well it is practised by the individual. Our model, therefore, demonstrates the maximal effect of the two different hand

washing schemes in this regard, but since we made this assumption for both, we do not expect it to impact our overall qualitative conclusions.

We have evaluated the effectiveness of hand washing on an individual's risk of infection mediated by the fomite-hand contamination route and did not take onward transmission into account.

There is limited literature on many parameters used in the model, which prevents us from making more precise quantitative conclusions. These include the probability of infection with contaminated hands, the survival of pathogens on contaminated hands and on surfaces and the infective dose. Furthermore, we modelled all infection events with the same rate of decay, i.e. the same probability of pathogen persistence on the hands. In reality, hand contamination events are likely to be heterogeneous with small droplets persisting only a short amount of time and heavy contamination with mucus decaying at a slower rate. In addition, we specifically focus on viral respiratory infections and assumed an exponential decay for the probability of viral persistence. While our model can be applied to all pathogens where hand hygiene is relevant for reducing respiratory tract infections, our results are only applicable for pathogens with a similar persistence behaviour. However, our model can be easily adapted if information on the persistence behaviour of specific pathogens is available.

## 5. Conclusion

To conclude, in this study we highlight the important considerations in hand hygiene behaviour to improve its effect in stopping the community spread of respiratory tract infections. Recommendations on hand hygiene should be tailored to the expected hand contamination rate and the half-life of virus persistence on hands.

Data accessibility. Data and code used in the analysis are publicly available: https://github.com/tm-pham/handhygiene_modelling.

Authors' contributions. M.P.: conceptualization, formal analysis, methodology, visualization, writing—original draft, writing—review and editing; M.Y.: conceptualization, investigation, writing—original draft, writing—review and editing; B.S.C.: conceptualization, methodology, project administration, supervision, writing—review and editing.

All authors gave final approval for publication and agreed to be held accountable for the work performed therein.

Conflict of interest declaration. We declare we have no competing interests.

Funding. This research was funded in whole, or in part, by the Wellcome Trust [220211]. T.M.P. was supported by the Society for Laboratory Automation and Screening, under award no.: SLAS_VS2020. Any opinions, findings and conclusions or recommendations expressed in this material are those of the author(s) and do not necessarily reflect those of the Society for Laboratory Automation and Screening. M.Y. is supported by the Singapore National Medical Research Council Research Fellowship (grant ref: NMRC/Fellowship/0051/2017).

Acknowledgements. We thank Ganna Rozhnova, Martin Bootsma and Mirjam Kretzschmar for helpful comments and discussions on the manuscript.

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
