## [Peer Review File · Proceedings. Mathematical, Physical, and Engineering Sciences]

Review History

RSPA-2021-0746.R0 (Original submission)

Review form: Referee 1

Is the manuscript an original and important contribution to its field?

Excellent

Is the paper of sufficient general interest?

Excellent

Is the overall quality of the paper suitable?

Good

Can the paper be shortened without overall detriment to the main message?

Yes

Do you think some of the material would be more appropriate as an electronic appendix?

Yes

Do you have any ethical concerns with this paper?

No

Recommendation?

Accept with minor revision (please list in comments)

Comments to the Author(s)

See report

Review form: Referee 2

Is the manuscript an original and important contribution to its field?

Good

Is the paper of sufficient general interest?

Good

Is the overall quality of the paper suitable?

Good

Can the paper be shortened without overall detriment to the main message?

Yes

Do you think some of the material would be more appropriate as an electronic appendix?

No

Do you have any ethical concerns with this paper?

No

Recommendation?

Accept with minor revision (please list in comments)

Comments to the Author(s)

The paper is a useful addition to the handwashing literature in that it models the potential effect of hand hygiene interventions according to timing and frequency of handwashing and survival of pathogen on hands. The conclusions have useful implications for the design of handwashing interventions in different settings to maximise impact.

As with all models a key challenge is the development of appropriate parameters to include in the model, The key insight is that if pathogens survive on the hand for only short periods then more frequent handwashing is needed to reduce risk compared to pathogens with longer survival on hands. This finding is highly robust to different values of parameters chosen.

The authors calibrate the model to a 10% secondary attack rate over 12 hours as the baseline attack rate - they say this is based on household secondary attack rates - however household secondary attack rates are driven by multiple routes of transmission not just surface to hand to mucous membrane - the direct droplet and aerosol part of the transmission are not amenable to interruption through hand hygiene - in this case, surely the model should be calibrated to the proportion of this secondary attack rate the authors think is not due to direct droplet or aerosol spread - could the authors clarify this point?

Another issue is that if viral survival on hands and surfaces is longer then this may lead to higher secondary attack rates but this does not seem to be factored into the model. By calibrating long and short survival viruses to the same baseline secondary attack rate the model is effectively presenting findings in terms of relative risk reductions rather than absolute risk reductions - if the absolute baseline risk is higher in longer survival viruses then the same relative risk reduction represents a bigger reduction in absolute risk for long-surviving viruses. This issue is not really factored into the results or policy implications. It might be usefully addressed by calibrating longer surviving viruses to a higher baseline attack rate than shorter surviving viruses.

The other main observation is that event prompted handwashing i.e. after the contamination event is more effective than regular handwashing at the same frequency.

This makes sense, however the authors have set the event prompted washing rate at 4 times per hour whereas the observational study they previously cite for touching of surfaces within an office is 112 times per hour. This really raises the question of whether event prompted hand hygiene is feasible. Perhaps this could be discussed.

The models focus on acquisition rather than spread of infection via the hand hygiene route. When the event in question is, for example blowing or wiping one's nose (catch it, kill it, bin it) then there is a clear event of interest. It would be interesting to model this or at least highlight in the discussion that that the strategies discussed do not take account of onward transmission from an infected subject.

The other important point that is perhaps underemphasised is that reducing the hand contamination rate is also effective in reducing risk. The authors mention the public health strategy of cleaning surfaces but do not mention reduction in surface touching as a strategy to reduce transmission.

Overall, I see this a very useful paper and my comments are aimed at allowing the authors to either refine their analysis or nuance their discussion, rather than seeing them as undermining their main conclusions.

Decision letter (RSPA-2021-0746.R0)

09-Dec-2021

Dear Mrs Pham

The Editor of Proceedings A has now received comments from referees on the above paper and would like you to revise it in accordance with their suggestions which can be found below (not including confidential reports to the Editor).

Please submit a copy of your revised paper within four weeks - if we do not hear from you within this time then it will be assumed that the paper has been withdrawn. In exceptional circumstances, extensions may be possible if agreed with the Editorial Office in advance.

Please note that it is the editorial policy of Proceedings A to offer authors one round of revision in which to address changes requested by referees. If the revisions are not considered satisfactory by the Editor, then the paper will be rejected, and not considered further for publication by the journal. In the event that the author chooses not to address a referee's comments, and no scientific

justification is included in their cover letter for this omission, it is at the discretion of the Editor whether to continue considering the manuscript.

To revise your manuscript, log into <http://mc.manuscriptcentral.com/prsa> and enter your Author Centre, where you will find your manuscript title listed under "Manuscripts with Decisions." Under "Actions," click on "Create a Revision." Your manuscript number has been appended to denote a revision.

You will be unable to make your revisions on the originally submitted version of the manuscript. Instead, revise your manuscript and upload a new version through your Author Centre.

When submitting your revised manuscript, you will be able to respond to the comments made by the referee(s) and upload a file "Response to Referees" in Step 1: "View and Respond to Decision Letter". Please provide a point-by-point response to the comments raised by the reviewers and the editor(s). A thorough response to these points will help us to assess your revision quickly. You can also upload a 'tracked changes' version either as part of the 'Response to reviews' or as a 'Main document'.

IMPORTANT: Your original files are available to you when you upload your revised manuscript. Please delete any unnecessary previous files before uploading your revised version.

When revising your paper please ensure that it remains under 28 pages long. In addition, any pages over 20 will be subject to a charge (£150 + VAT (where applicable) per page). Your paper has been ESTIMATED to be 18 pages.

Open Access

You are invited to opt for open access, our author pays publishing model. Payment of open access fees will enable your article to be made freely available via the Royal Society website as soon as it is ready for publication. For more information about open access please visit <https://royalsociety.org/journals/authors/open-access/>. The open access fee for this journal is £1700/\$2380/€2040 per article. VAT will be charged where applicable. Please note that if the corresponding author is at an institution that is part of a Read and Publishing deal you are required to select this option. See <https://royalsociety.org/journals/librarians/purchasing/read-and-publish/read-publish-agreements/> for further details.

Once again, thank you for submitting your manuscript to Proc. R. Soc. A and I look forward to receiving your revision. If you have any questions at all, please do not hesitate to get in touch.

Yours sincerely
Raminder Shergill
proceedingsa@royalsociety.org

on behalf of
Professor Matjaz Perc
Board Member
Proceedings A

Reviewer(s)' Comments to Author:
Referee: 1
Comments to the Author(s)
See report

Referee: 2

Comments to the Author(s)

The paper is a useful addition to the handwashing literature in that it models the potential effect of hand hygiene interventions according to timing and frequency of handwashing and survival of pathogen on hands. The conclusions have useful implications for the design of handwashing interventions in different settings to maximise impact.

As with all models a key challenge is the development of appropriate parameters to include in the model, The key insight is that if pathogens survive on the hand for only short periods then more frequent handwashing is needed to reduce risk compared to pathogens with longer survival on hands. This finding is highly robust to different values of parameters chosen.

The authors calibrate the model to a 10% secondary attack rate over 12 hours as the baseline attack rate - they say this is based on household secondary attack rates - however household secondary attack rates are driven by multiple routes of transmission not just surface to hand to mucous membrane - the direct droplet and aerosol part of the transmission are not amenable to interruption through hand hygiene - in this case, surely the model should be calibrated to the proportion of this secondary attack rate the authors think is not due due to direct droplet or aerosol spread - could the authors clarify this point?

Another issue is that if viral survival on hands and surfaces is longer then this may lead to higher secondary attack rates but this does not seem to be factored into the model. By calibrating long and short survival viruses to the same baseline secondary attack rate the model is effectively presenting findings in terms of relative risk reductions rather than absolute risk reductions - if the absolute baseline risk is higher in longer survival viruses then the same relative risk reduction represents a bigger reduction in absolute risk for long-surviving viruses. This issue is not really factored into the results or policy implications. It might be usefully addressed by calibrating longer surviving viruses to a higher baseline attack rate than shorter surviving viruses.

The other main observation is that event prompted handwashing i.e. after the contamination event is more effective than regular handwashing at the same frequency. This makes sense, however the authors have set the event prompted washing rate at 4 times per hour whereas the observational study they previously cite for touching of surfaces within an office is 112 times per hour. This really raises the question of whether event prompted hand hygiene is feasible. Perhaps this could be discussed.

The models focus on acquisition rather than spread of infection via the hand hygiene route. When the event in question is, for example blowing or wiping one's nose (catch it, kill it, bin it) then there is a clear event of interest. It would be interesting to model this or at least highlight in the discussion that that the strategies discussed do not take account of onward transmission from an infected subject.

The other important point that is perhaps underemphasised is that reducing the hand contamination rate is also effective in reducing risk. The authors mention the public health strategy of cleaning surfaces but do not mention reduction in surface touching as a strategy to reduce transmission.

Overall, I see this a very useful paper and my comments are aimed at allowing the authors to either refine their analysis or nuance their discussion, rather than seeing them as undermining their main conclusions.

Author's Response to Decision Letter for (RSPA-2021-0746.R0)

See Appendix B.

RSPA-2021-0746.R1 (Revision)

Review form: Referee 1

Is the manuscript an original and important contribution to its field?

Excellent

Is the paper of sufficient general interest?

Excellent

Is the overall quality of the paper suitable?

Good

Can the paper be shortened without overall detriment to the main message?

Yes

Do you think some of the material would be more appropriate as an electronic appendix?

No

Do you have any ethical concerns with this paper?

No

Recommendation?

Accept as is

Comments to the Author(s)

I believe that the authors have successfully addressed all my comments, and I am happy to recommend publication as it is.

Decision letter (RSPA-2021-0746.R1)

07-Apr-2022

Dear Mrs Pham

I am pleased to inform you that your manuscript entitled "The potential impact of intensified community hand hygiene interventions on respiratory tract infections: a modelling study" has been accepted in its final form for publication in Proceedings A.

Our Production Office will be in contact with you in due course. You can expect to receive a proof of your article soon. Please contact the office to let us know if you are likely to be away from e-

mail in the near future. If you do not notify us and comments are not received within 5 days of sending the proof, we may publish the paper as it stands.

As a reminder, you have provided the following 'Data accessibility statement' (if applicable). Please remember to make any data sets live prior to publication, and update any links as needed when you receive a proof to check. It is good practice to also add data sets to your reference list. Statement (if applicable): The code reproducing the results of this study is available at https://github.com/tm-pham/handhygiene_modelling.

Open access

You are invited to opt for open access, our author pays publishing model. Payment of open access fees will enable your article to be made freely available via the Royal Society website as soon as it is ready for publication. For more information about open access please visit <https://royalsociety.org/journals/authors/which-journal/open-access/>. The open access fee for this journal is £1700/\$2380/€2040 per article. VAT will be charged where applicable.

Note that if you have opted for open access then payment will be required before the article is published – payment instructions will follow shortly.

If you wish to opt for open access then please inform the editorial office (proceedingsa@royalsociety.org) as soon as possible.

Your article has been estimated as being 14 pages long. Our Production Office will inform you of the exact length at the proof stage.

Proceedings A levies charges for articles which exceed 20 printed pages. (based upon approximately 540 words or 2 figures per page). Articles exceeding this limit will incur page charges of £150 per page or part page, plus VAT (where applicable).

Under the terms of our licence to publish you may post the author generated postprint (ie. your accepted version not the final typeset version) of your manuscript at any time and this can be made freely available. Postprints can be deposited on a personal or institutional website, or a recognised server/repository. Please note however, that the reporting of postprints is subject to a media embargo, and that the status the manuscript should be made clear. Upon publication of the definitive version on the publisher's site, full details and a link should be added.

You can cite the article in advance of publication using its DOI. The DOI will take the form: 10.1098/rspa.XXXX.YYYY, where XXXX and YYYY are the last 8 digits of your manuscript number (eg. if your manuscript number is RSPA-2017-1234 the DOI would be 10.1098/rspa.2017.1234).

For tips on promoting your accepted paper see our blog post: <https://royalsociety.org/blog/2020/07/promoting-your-latest-paper-and-tracking-your-results/>

On behalf of the Editor of Proceedings A, we look forward to your continued contributions to the Journal.

Sincerely,
Raminder Shergill
proceedingsa@royalsociety.org

on behalf of
Professor Matjaz Perc
Board Member
Proceedings A

Reviewer(s)' Comments to Author:

Referee: 1

Comments to the Author(s)

I believe that the authors have successfully addressed all my comments, and I am happy to recommend publication as it is.

Manuscript Number: RSPA-2021-0746

Title: The potential impact of intensified community hand hygiene interventions on respiratory tract infections: a modelling study.

In this work, authors aim to quantify the impact of interventions related to hand hygiene on exposure to respiratory tract infections. To do this, they consider a relatively simple mathematical model which considers exposure for a single individual through the fomite route. In particular, the model represents: (i) contamination of hand(s) via contact with contaminated surfaces, but where the contamination of these surfaces is not explicitly modelled; (ii) two different hand-hygiene regimes based on regular hand washing vs hand washing prompted after surface contact with some delay; (iii) persistence of virus on hands in terms of an exponential decay; and (iv) exposure via contacts of the hand with membranes.

I think this is a nice contribution to the field, and I think it should be published after authors address the comments below. The role played by the fomite transmission route has not been very extensively analysed yet, and this type of approach strikes a nice balance between model complexity, feasibility of analysis and interpretability of results. Still, I have a number of comments/questions/concerns that I believe the authors should address.

Major comments, mainly related to model assumptions:

- It is not clear (not to me), from the description in Figures 1 and 2 and within the text, what happens when a sequence of hand contamination events occur. Would the "Probability of infection" curve in Figure 1 go back to its baseline peak original value if another hand contamination event occurs before hand washing?
- λ_c is the rate of the Poisson process for the number of hand contamination events. This parameter will effectively depend on: (a) the rate at which the individual touches surfaces; and (b) the percentage of surfaces that are contaminated in that given environment at any given time. One could just vary this parameter to account for heterogeneity in (a) (across individuals and situations in real life). However, more thought needs to be made for (b). If the authors are considering as a major factor in the study the "persistence of the pathogen on hands", it is to be expected that the persistence (eg half-life) on hands and on surfaces would be highly correlated for many pathogens. This means that pathogens that survive for longer on hands would likely survive also for longer on surfaces, and then a fair comparison would imply considering larger values for λ_c for these pathogens (i.e. when $T_{1/2}$ is large)? I think some numerical results should focus on this, and some analysis would be needed for the conclusions in the paper to be robust.
- It is a standard approach in Quantitative Microbial Risk Assessment (QMRA) to explicitly model the level of contamination on hands and surfaces (e.g. PFU/cm^2). The authors have instead followed a simpler approach, which has limitations but leads to a simple model which provides useful insights. Still, the fact that the level of contamination (at least) on hands is not explicitly modelled is a very important limitation of the study, which is currently not mentioned. In general, and related to this, I got the feeling that some relevant literature is missing in the manuscript:
 - QMRA modelling for fomite transmission, where the level of contamination on hands and surfaces is explicitly modelled, and hand washing has been sometimes addressed. See for example works by Yuguo Li et al., Tim Julian et al. or King & Wilson et al., among others.
 - When the authors discuss about the relative contribution of each transmission routes in Page 10, they could point the reader to QMRA studies where these relative contributions have been

quantified. See, e.g., Lei et al. (for an air cabin), Azimi et al. (for the Diamond princess cruise) or Jones (2020) for a hospital setting.

The fact that the level of contamination on hands is not explicitly modelled leads to a number of relatively strong assumptions in the model, which relate to my following points.

- ϵ is the probability of infection per hand-mucosa contact. In a real scenario, this probability would depend on: (a) the level of contamination on hands (PFU/cm^2); (b) the hand area involved on the hand-mucosa contact; (c) the number of contacts in a relatively short period of time, leading to an accumulation of dose; and (d) the infectivity of the pathogen and susceptibility of the host (eg in terms of a dose-response curve). Merging these factors into a single fixed parameter is a limitation of the study which should be highlighted. In particular,
 - ϵ is unlikely to be independent across sequential contacts. If ϵ is large due to high hand contamination levels in a given contact, it will likely be large in the next contact for the same reason. Some of this is partially accounted for through the exponential decay of virus on hands, but not fully (eg very large initial contamination levels on hands would lead to a very large probability of infection for particular dose-reponse curves, regardless of the decay happening).
 - Doses can accumulate on membranes across sequential contacts in the short-term, leading to an accumulation of risk (i.e. risk for the cumulative dose).
 - The approach followed by the authors to set ϵ (e.g. Figure S4) is not fully clear to me. Why would ϵ depend on the half-life? I understand that they are trying to make things comparable by fixing the cumulative probability of transmission, but is this a fair comparison? A particular pathogen would have a given half-life for persistence on hands, and a given ϵ based on (a)-(d) above. Does it make sense to *force* the two pathogens with different half-lives to be “equally infectious” by accordingly modifying ϵ ? This might be affecting some of the conclusions. But I might have misunderstood something here.
- Authors should clarify when they talk about infectious virus vs viral RNA. In Figure S1, they mention infectious virus, but in Section 2(d) they mention a study [15] which detected virus (I think here it is just RNA) on 53% of surfaces. If it is RNA, I imagine this might lead to an overestimate of contaminated surfaces for the purposes of infection transmission modelling, since I believe the ratio between viral RNA and infectious virus can be as small as 1:1000, and not all those 53% of surfaces would be “infectious” (or infectious enough to contaminate the hand enough to cause infection via ϵ later on).
- Authors choose a cumulative probability of infection of 10% over a period of 12h based on attack rates for Influenza from [10-13]. Are these quantified rates for fomite transmission? or in general regardless of the route? If these are general, then 10% in 12h might be a significant over estimate of infection risk via the fomite route.
- Some sensitivity analysis for λ_f is needed, I believe, unless I have missed it. The face-touching rate can vary across individuals, ages (e.g. kids vs adults), but also depending on environmental conditions (public vs private spaces; facemasks etc). This parameter might have a significant impact on some of the exposure predictions, where hand hygiene might not be effective when λ_f is large, regardless of the hand washing frequency strategy.

Minor comments:

- For consistency, authors might want to use the same symbol (X) for transmission blocked in Figures 1 and 2.

- Authors state “*We assumed that when hand washing is performed... $P(t_i)$ is reduced to zero*”. It is probably a sensible assumption given high efficacy of hand washing, but authors might want to support this statement with some references/arguments. Is this implicitly assuming not only that the sanitizer has large efficacy, but also that the hand-washing is performed correctly?
- Computed instead of cmputed in Figure S4 caption.
- The Github link in the Data Accessibility statement is broken, probably because of the index h. In this repository, the authors might want to add some extra description of the codes in the README file.
- Please explain how the cumulative probability of infection expression in line 55, Page 12, is obtained.

Appendix B

Dear Raminder Shergill and Professor Matjaz Perc,

Herewith we would like to resubmit the revised manuscript entitled "The potential impact of intensified community hand hygiene interventions on respiratory tract infections: a modelling study" (RSPA-2021-0746) for your consideration for publication in *Proceedings of the Royal Society A*.

We would like to thank the editors and the reviewers for the valuable comments and suggestions, and the opportunity to revise our manuscript. We have carefully revised the manuscript with changes marked in blue and a point-by-point reply to the reviewers.

Sincerely,

Thi Mui Pham

on behalf of all authors

Reviewer 1

In this work, authors aim to quantify the impact of interventions related to hand hygiene on exposure to respiratory tract infections. To do this, they consider a relatively simple mathematical model which considers exposure for a single individual through the fomite route. In particular, the model represents: (i) contamination of hand(s) via contact with contaminated surfaces, but where the contamination of these surfaces is not explicitly modelled; (ii) two different hand-hygiene regimes based on regular hand washing vs hand washing prompted after surface contact with some delay; (iii) persistence of virus on hands in terms of an exponential decay; and (iv) exposure via contacts of the hand with membranes.

I think this is a nice contribution to the field, and I think it should be published after authors address the comments below. The role played by the fomite transmission route has not been very extensively analysed yet, and this type of approach strikes a nice balance between model complexity, feasibility of analysis and interpretability of results. Still, I have a number of comments/questions/concerns that I believe the authors should address.

Major comments, mainly related to model assumptions:

1. It is not clear (not to me), from the description in Figures 1 and 2 and within the text, what happens when a sequence of hand contamination events occur. Would the "Probability of infection" curve in Figure 1 go back to its baseline peak original value if another hand contamination event occurs before hand washing?

Response: Thank you for pointing out that our manuscript did not address the consequences of a sequence of hand contamination events. We indeed assume in our model that, in case of a sequence of hand contamination events, the probability of viral persistence is reset to one after each hand contamination event while the probability of transmission per face-touching contact ϵ is kept the same. The probability infection is, thus, reset to its baseline value. We, therefore, assume no accumulation of infection risk. In addition, we also assume that the functional shape of this probability is always the same. Our model, therefore, might underestimate the effect of hand washing in reducing the cumulative probability of infection. We have added the following sentence to the Methods section ("Mathematical model", page 4-5):

"In case of a sequence of hand contamination events, we assumed that the probability $P(t)$ is reset to its baseline value of one and that the transmission probability ϵ is kept the same. As such, the probability $P(t)$ is assumed to always have the same functional shape even after

sequential hand contamination events. We, therefore, do not account for accumulation of infection risks neither for heterogeneity of hand contamination events."

We also added the following paragraph to the Discussion section (page 10):

"We assumed no accumulation of the probability of infection for a sequence of hand contamination events and did not account for heterogeneity of hand contamination events. Instead, the probability of viral persistence is reset to its baseline value of one and the transmission probability ϵ is kept equal for consecutive hand contamination events. Our model, therefore, might underestimate the reduction in the probability of infection induced by hand washing. However, since we do not expect this to affect the hand washing schemes differently, we do not expect this to change our qualitative conclusions."

2. λ_c is the rate of the Poisson process for the number of hand contamination events. This parameter will effectively depend on: (a) the rate at which the individual touches surfaces; and (b) the percentage of surfaces that are contaminated in that given environment at any given time. One could just vary this parameter to account for heterogeneity in (a) (across individuals and situations in real life). However, more thought needs to be made for (b). If the authors are considering as a major factor in the study the "persistence of the pathogen on hands", it is to be expected that the persistence (eg half-life) on hands and on surfaces would be highly correlated for many pathogens. This means that pathogens that survive for longer on hands would likely survive also for longer on surfaces, and then a fair comparison would imply considering larger values for λ_c for these pathogens (i.e. when $T_{1/2}$ is large)? I think some numerical results should focus on this, and some analysis would be needed for the conclusions in the paper to be robust.

Response: We agree that λ_c depends both on (a) and (b) and we agree that there might be a correlation between the persistence on hands and the hand contamination rate λ_c . However, it's not clear what exactly this relationship would look like as pathogens may behave very differently on human tissue than on artificial surfaces. In addition, this can vary across pathogens and depend on the surface type and environmental conditions. As we will discuss in our response to 4. iii), our study highlights that for a new pathogen with observed secondary attack rates (e.g., through household studies) we would need to find out about the half-life on contaminated hands to make statements about how transmission would be expected to change with increasing hand hygiene. For the analyses in the main text, we fixed the baseline probability of infection and the hand contamination rate, assuming that these parameters could be determined through other (household) studies. If indeed, there is a

correlation between the half-life and the hand contamination rate, our analyses need to be extended and we have explored this in the supplementary material (page 34ff. "Correlation between half-life of viral persistence on hands and hand contamination rate"). The results in Figure S20 and Figure S21 show that the combination of half-life of viral persistence and hand contamination rate is an important determinant for assessing the effect of hand washing on the probability of infection. Given that it is unclear what form this relationship takes, that the relationship may vary across pathogens and settings, and that it is tangential to the main focus of our manuscript, we prefer not to introduce further explicit assumptions about the precise form of this relationship. We have added the following text to the Discussion section (page 10) to address this point:

"Further analyses included in our supplementary material show that the relationship between the hand contamination rate and viral survival on hands is an important determinant for evaluating the effect of hand washing on the risk of fomite-mediated infection. This relationship remains underexplored and further investigations are needed for each pathogen in question to reliably estimate the potential impact of hand washing."

3. It is a standard approach in Quantitative Microbial Risk Assessment (QMRA) to explicitly model the level of contamination on hands and surfaces (e.g. PFU/cm²). The authors have instead followed a simpler approach, which has limitations but leads to a simple model which provides useful insights. Still, the fact that the level of contamination (at least) on hands is not explicitly modelled is a very important limitation of the study, which is currently not mentioned. In general, and related to this, I got the feeling that some relevant literature is missing in the manuscript:

- QMRA modelling for fomite transmission, where the level of contamination on hands and surfaces is explicitly modelled, and hand washing has been sometimes addressed. See for example works by Yuguo Li et al., Tim Julian et al. or King & Wilson et al., among others.
- When the authors discuss about the relative contribution of each transmission routes in Page 10, they could point the reader to QMRA studies where these relative contributions have been quantified. See, e.g., Lei et al. (for an air cabin), Azimi et al. (for the Diamond princess cruise) or Jones (2020) for a hospital setting.

Response: Thank you for pointing out the missing literature in the manuscript. We have incorporated many of the suggested articles along with additional literature on that topic in the introduction (page 2) and the discussion (page 10). In particular, we have also added literature on the relative contribution of fomite-mediated transmission routes for influenza in

the Methods section (see "Parameters", page 5). We agree that our decision to neglect a dose-response relationship and the level of contamination of hands is a limitation of our study. However, we think that our simpler model provides a sufficient framework to assess the effect of the two hand washing schemes that we considered.

4. The fact that the level of contamination on hands is not explicitly modelled leads to a number of relatively strong assumptions in the model, which relate to my following points.
 - ϵ is the probability of infection per hand-mucosa contact. In a real scenario, this probability would depend on: (a) the level of contamination on hands (PFU=cm²); (b) the hand area involved on the hand-mucosa contact; (c) the number of contacts in a relatively short period of time, leading to an accumulation of dose; and (d) the infectivity of the pathogen and susceptibility of the host (eg in terms of a dose-response curve). Merging these factors into a single fixed parameter is a limitation of the study which should be highlighted.

Response: We agree with the reviewer that our decision to approach this problem using a very simple model has neglected a lot of real-world biological complexity. In line with the reviewer's suggestion, we now highlight some of this neglected complexity in the discussion. In addition, we have added the following sentence to the Methods section (page 4): *"This parameter combines several factors such as the level of contamination on hands, the infectivity of the virus, and the susceptibility of the host. We do not explicitly model a dose-response relationship and thus, ϵ represents the average probability of infection per hand-mucosa contact."*

In particular,

- i. it is unlikely to be independent across sequential contacts. If ϵ is large due to high hand contamination levels in a given contact, it will likely be large in the next contact for the same reason. Some of this is partially accounted for through the exponential decay of virus on hands, but not fully (eg very large initial contamination levels on hands would lead to a very large probability of infection for particular dose-response curves, regardless of the decay happening).

Response: We assumed that ϵ is fixed and constant for all infection events. As such, each self-infection event is maximally correlated as it is assumed to be the same. However, as we did not account for different levels of hand

contamination, we did not incorporate a dose-response relationship. We now acknowledge this limitation in the Discussion section (page 10):

"Thirdly, we did not account for variation and likely autocorrelation in the probability of infection per hand-mucosa contact or for non-homogeneity in the rate of hand contamination. While incorporating such factors would have some effect on precise quantitative results reported, they would not affect the broad qualitative conclusions."

- ii. Doses can accumulate on membranes across sequential contacts in the short-term, leading to an accumulation of risk (i.e. risk for the cumulative dose).

Response: We acknowledge that we neglected the accumulation of infection risk through sequential contacts and have added this assumption to the Methods section as well as discuss this limitation in the Discussion section (page 10 in manuscript, see also our response to Comment 1):

"We assumed no accumulation of the probability of infection for a sequence of hand contamination events and did not account for heterogeneity of hand contamination events. Instead, the probability of viral persistence is reset to its baseline value of one and the transmission probability ϵ is kept equal for consecutive hand contamination events. Our model, therefore, might underestimate the reduction in the probability of infection induced by hand washing. However, since we do not expect this to affect the hand washing schemes differently, we do not expect this to change our qualitative conclusions."

- iii. The approach followed by the authors to set ϵ (e.g. Figure S4) is not fully clear to me. Why would depend on the half-life? I understand that they are trying to make things comparable by fixing the cumulative probability of transmission, but is this a fair comparison? A particular pathogen would have a given half-life for persistence on hands, and a given based on (a)-(d) above. Does it make sense to force the two pathogens with different half-lives to be equally infectious" by accordingly modifying ϵ ? This might be affecting some of the conclusions. But I might have misunderstood something here.

Response: Indeed the reviewer is correct: in Figure 3 we are interested in comparing the impact of hand washing frequency on pathogens of equal

transmissibility but with different half-lives, and this motivated our decision to vary ϵ to achieve the same overall transmission probability for the different pathogens. The purpose of this was to highlight the impact of the half-life (which is often poorly quantified) on the effectiveness of hand hygiene, thus highlighting that for a new pathogen of known transmissibility we would need to find out about the half-life on contaminated hands to make statements about how transmission would be expected to change with increasing hand hygiene. We certainly didn't mean to imply that increased persistence of a pathogen on hands would not increase transmissibility. To address this potential confusion we have now added the following text to the Methods section ("Parameter", page 5):

"The probability of transmission per face-touching event was constrained to meet a fixed probability of infection to reflect the fact that we are interested in how our beliefs about how the potential impact of enhanced hand hygiene for a pathogen of known transmissibility will vary according to what we know or believe about its survival on hands."

5. Authors should clarify when they talk about infectious virus vs viral RNA. In Figure S1, they mention infectious virus, but in Section 2(d) they mention a study [15] which detected virus (I think here it is just RNA) on 53% of surfaces. If it is RNA, I imagine this might lead to an overestimate of contaminated surfaces for the purposes of infection transmission modelling, since I believe the ratio between viral RNA and infectious virus can be as small as 1:1000, and not all those 53% of surfaces would be "infectious" (or infectious enough to contaminate the hand enough to cause infection via later on).

Response: Indeed, in the paper by Boone et al (2004) that we cite viral viability was not assessed. Assuming 100% of detected RNA is from viable virus, we used this paper to inform the upper bound for the hand contamination event rate. However, we acknowledge that a direct translation from detected RNA to viable virus is likely unrealistic and should be seen as a theoretical upper bound. We have added the following sentence to the Methods section (see "Parameters", page 5) to highlight this assumption:

"Note that $\lambda = 60 \text{ hour}^{-1}$ is based on a RNA to viable virus ratio of 1:1 and should be seen as a theoretical upper bound as in practice, this ratio is likely much smaller."

In our main analysis, we used a much lower hand contamination rate of 4 events per hour. We performed sensitivity analyses for hand contamination rates of 1, 20 and 60 (page 28ff. in

supplementary material). In fact, for very high hand contamination rates the effect of hand washing on the probability of infection is small and fixed-time hand washing may become more efficient (reduction in probability of infection when compared to similar average number of hand washing events per hour) and event-prompted hand washing with short delays may become infeasible. For low hand contamination rates, the conclusions from our main analysis remain unchanged.

6. Authors choose a cumulative probability of infection of 10% over a period of 12h based on attack rates for Influenza from [10-13]. Are these quantified rates for fomite transmission? or in general regardless of the route? If these are general, then 10% in 12h might be a significant overestimate of infection risk via the fomite route.

Response: The secondary attack rates from the cited studies are in fact not adjusted for the fomite transmission route. Hence, the actual cumulative probability of infection might be lower than what has been reported. Unfortunately, little is known about the relative contribution of transmission routes for respiratory viruses and may vary for different pathogens (as discussed in the Discussion section, page 10ff.). As secondary attack rates also varied across studies and across age groups (values range from 8% to 50%), we performed additional sensitivity analyses with respect to this parameter. We have now added sensitivity analyses for the cumulative probability of infection of 0.01%, 1%, and 5% to the supplementary material (additional to the ones for 30% and 50%). Together with the main analyses, we now cover a broad spectrum of values and show in the supplementary material (page 18ff.) that our conclusions remain unchanged when this parameter is varied (and all others are fixed).

7. Some sensitivity analysis for λ_f is needed, I believe, unless I have missed it. The face-touching rate can vary across individuals, ages (e.g. kids vs adults), but also depending on environmental conditions (public vs private spaces; facemasks etc). This parameter might have a significant impact on some of the exposure predictions, where hand hygiene might not be effective when λ_f is large, regardless of the handwashing frequency strategy.

Response: We agree with the reviewer that λ_f might depend on characteristics and behaviour of the individual as well as the setting. Frequencies for hand-face contacts depend on the area of the face that is considered and vary considerably between studies (15.7 reported in Nicas et al (2008), 27.7 reported in Lewis et al (2020)). Wilson et al (2020) showed that this rate depends also on the activity with generally higher contact rates for eating vs non-eating activities as well as the area of the face (lower contact frequencies for mouth, eyes, and nose vs all other face areas). Informed by these additional studies, we

added sensitivity analyses for λ_f of 5, 20 and 50 times per hour to the supplementary material (page 21ff.). Note that in our main analysis we used $\lambda_f=10$ based on the study of Kwok et al and adjusted for the face touches that included mucosal areas as face touching events might not necessarily directly translate to infection events. Our sensitivity analyses showed that the main conclusions did not change if λ_f was varied but all other parameters remained fixed.

Minor comments:

8. For consistency, authors might want to use the same symbol (X) for transmission blocked in Figures 1 and 2.

Response: Thank you for this suggestion. We have changed the square to the symbol X in Figure 2.

9. Authors state "We assumed that when hand washing is performed... P(ti) is reduced to zero". It is probably a sensible assumption given high efficacy of hand washing, but authors might want to support this statement with some references/arguments. Is this implicitly assuming not only that the sanitizer has large efficacy, but also that the hand-washing is performed correctly?

Response: We have added a reference by Grayson et al (2009) and Larson et al (2012) to support this assumption. This indeed implicitly assumes a high efficacy of hand washing itself but also that it is performed correctly. We agree that this assumption might be optimistic and our results, therefore, show the maximum effect of hand washing on the probability of infection. We have added this limitation to the Discussion section (page 10):

"Fourthly, we assumed that hand washing reduced the probability of persistence on hands to zero and hence that hand washing is maximally efficacious in removing the virus from hands. Although a high efficacy of hand hygiene with soap and water and alcohol-based hand rubs has been demonstrated [28, 29], its real world effectiveness depends on how well it is practised by the individual. Our model, therefore, demonstrates the maximal effect of the two different hand washing schemes in this regard, but since we made this assumption for both, we do not expect it to impact our overall qualitative conclusions."

10. Computed instead of ccomputed in Figure S4 caption.

Response: We have corrected this spelling mistake.

11. The Github link in the Data Accessibility statement is broken, probably because of the index h. In this repository, the authors might want to add some extra descriptions of the codes in the README file.

Response: We had changed the name of the Github repository to https://github.com/tm-pham/handhygiene_modelling but the link in the manuscript was not updated. We have corrected this mistake and added some descriptions of the code in the README file.

12. Please explain how the cumulative probability of infection expression in line 55, Page 12, is obtained.

Response: We added the following text to further explain this expression on page 14:

"The cumulative probability of infection over the time period T is given by

$$1 - \prod_{i=1}^F 1 - \epsilon P(t_i)$$

and can be approximated by

$$1 - e^{-\sum_{i=1}^F \epsilon P(t_i)}$$

This represents the complement of the probability that the transmission events 1 ... F do not lead to infection."

Reviewer 2

The paper is a useful addition to the handwashing literature in that it models the potential effect of hand hygiene interventions according to timing and frequency of handwashing and survival of pathogen on hands. The conclusions have useful implications for the design of handwashing interventions in different settings to maximise impact.

As with all models a key challenge is the development of appropriate parameters to include in the model, The key insight is that if pathogens survive on the hand for only short periods then more frequent handwashing is needed to reduce risk compared to pathogens with longer survival on hands. This finding is highly robust to different values of parameters chosen.

1. The authors calibrate the model to a 10% secondary attack rate over 12 hours as the baseline attack rate - they say this is based on household secondary attack rates - however household secondary attack rates are driven by multiple routes of transmission not just surface to hand to mucous membrane - the direct droplet and aerosol part of the transmission are not amenable to interruption through hand hygiene - in this case, surely the model should be

calibrated to the proportion of this secondary attack rate the authors think is not due to direct droplet or aerosol spread - could the authors clarify this point?

Response: Indeed, secondary attack rates in household studies represent multiple routes of transmission. The relative contribution of each of these routes depends on many variables such as environmental factors (e.g., humidity and temperature), the setting (healthcare or community setting, number of people in a room), the pathogen, and the host. However, to date, the relative importance of transmission routes stays inconclusive for respiratory viruses (see, e.g., Kutter et al (2018), Killingley et al (2013)) and varies significantly between different viruses. For influenza and SARS-CoV-2, the relative contribution for contact transmission varies between 1% and 50%. In addition, there is a large variation among secondary attack rates even for the same pathogen but for different age categories. In particular, higher secondary attack rates were reported for children (under 16) in comparison with adults for influenza-like illness and acute respiratory illness (25.4% and 42.4% compared to 7.6% and 17.2%, respectively). To account for this variability, we have added more scenarios with different probabilities of infection to our analysis. Specifically, we have now included sensitivity analyses with probabilities of infection of 0.1%, 1%, and 5%. Our overall conclusions did not change for all values that we explored. We also added a paragraph to the Discussion section:

"For example, in a two-route transmission model, the fomite route was estimated to contribute 6% to the infections caused by influenza in a nosocomial outbreak in Hong Kong [37]. For SARS-CoV-2,

Azimi and colleagues developed a mechanistic transmission modeling framework with multiple transmission routes and used detailed information available from the Diamond Princess cruise ship outbreak [38]. The fomite-mediated mode was estimated to contribute to 21% (median) of infected cases aboard the ship. In a healthcare setting, the contribution of contact transmission was estimated to contribute 8.2% (0.0, 0.37%) to the overall infection risk for SARS-CoV-2 [39]. In a comparative analysis of outbreaks of influenza H1N1, SARS-CoV, and norovirus in air cabins, the fomite route was estimated to play the dominant role for SARS-CoV (50%, 95% CI: 48%-53%) while its contribution was minor for H1N1 (less than 1%) [40]. When other routes are considered, the baseline probability of infection needs to be adapted with lower values leading to a lower absolute effect of hand hygiene."

2. Another issue is that if viral survival on hands and surfaces is longer then this may lead to higher secondary attack rates but this does not seem to be factored into the model. By calibrating long and short survival viruses to the same baseline secondary attack rate the

model is effectively presenting findings in terms of relative risk reductions rather than absolute risk reductions - if the absolute baseline risk is higher in longer survival viruses then the same relative risk reduction represents a bigger reduction in absolute risk for long-surviving viruses. This issue is not really factored into the results or policy implications. It might be usefully addressed by calibrating longer surviving viruses to a higher baseline attack rate than shorter surviving viruses.

Response: We agree with the reviewer that we have not taken the correlation between viral survival on hands and increased transmissibility into account. However, we wanted to highlight that for a new pathogen of known transmissibility (e.g., observed as secondary attack rates in household studies) we would need to find out about the half-life on contaminated hands to make statements about how transmission would be expected to change with increasing hand hygiene. Thus, we deliberately fixed the cumulative probability of infection to study the impact of the half-life of viral persistence (which is often poorly understood and quantified) on the effectiveness of hand washing. We show that the half-life has a large impact and it is, therefore, important to quantify this parameter to estimate the effectiveness of hand hygiene to reduce infection risk mediated by fomites.

To address this potential confusion, we have added the following text to the Methods section (page 5, "Parameters"): *"The probability of transmission per face-touching event was constrained to meet a fixed probability of infection to reflect the fact that we are interested in how our beliefs about how the potential impact of enhanced hand hygiene for a pathogen of known transmissibility will vary according to what we know or believe about its survival on hands."*

3. The other main observation is that event prompted handwashing i.e. after the contamination event is more effective than regular handwashing at the same frequency. This makes sense, however the authors have set the event prompted washing rate at 4 times per hour whereas the observational study they previously cite for touching of surfaces within an office is 112 times per hour [ref]. This really raises the question of whether event prompted hand hygiene is feasible. Perhaps this could be discussed.

Response: We agree with the reviewer that, if hand contamination rates are very frequent, an event-prompted hand washing strategy is likely not feasible. However, although the surface touch rate in an office might be as high as 112 times per hour, not all of these touches might be hand contamination events. In fact, we cited another study where they detected RNA virus on 53% in households (Boone et al, 2005). In this study, they did not test whether the virus was viable. We, therefore, think that the hand contamination rate is much

smaller than the surface touching rate. Due to the uncertainty of this parameter, we performed sensitivity analyses for the hand contamination rate of 1, 10, 20, and 60 (see supplementary material, page 24ff.). We agree that for high hand contamination rates, event-prompted hand washing with a short delay becomes infeasible (similar to fixed-time hand washing with short time intervals). We have refined this point in the Discussion of our manuscript (page 9):

"Furthermore, in the second case, where hands become contaminated very frequently, a substantial reduction in the probability of infection is unlikely to be attained unless hand washing frequency is increased drastically, i.e., every one to five minutes. Because hand washing at such a high rate is not practical (neither for fixed-time nor event-prompted hand washing), the recommendation in this scenario is to regularly clean the environment (such as surfaces), to reduce the rate of surface touching (if possible), and/or isolate infected individuals to reduce hand contamination events."

4. The models focus on acquisition rather than spread of infection via the hand hygiene route. When the event in question is, for example blowing or wiping one's nose (catch it, kill it, bin it) then there is a clear event of interest. It would be interesting to model this or at least highlight in the discussion that the strategies discussed do not take account of onward transmission from an infected subject.

Response: We agree with the reviewer that it would be interesting to explore onward transmission from an infected individual and its interdependence with hand washing strategies. However, this would require a full transmission model and is out of the scope of this study. Here, we focus on the effect of hand washing on the individual's risk of infection rather than its effect on a population level. We have added this limitation to the Discussion section of our manuscript (page 10):

"We have evaluated the effectiveness of hand washing on an individual's risk of infection mediated by the fomite-hand contamination route and did not take onward transmission into account."

5. The other important point that is perhaps underemphasised is that reducing the hand contamination rate is also effective in reducing risk. The authors mention the public health strategy of cleaning surfaces but do not mention reduction in surface touching as a strategy to reduce transmission.

Response: We agree with the reviewer that the reduction of the hand contamination rate represents an alternative strategy to reduce fomite-mediated transmission. We have added this point in the respective paragraph in our discussion:

"Furthermore, in the second case, where hands become contaminated very frequently, a substantial reduction in the probability of infection is unlikely to be attained unless hand washing frequency is increased drastically, i.e., every one to five minutes. Because hand washing at such a high rate is not practical (neither for fixed-time nor event-prompted hand washing), the recommendation in this scenario is to regularly clean the environment (such as surfaces), to reduce the rate of surface touching (if possible), and/or isolate infected individuals to reduce hand contamination events."

In addition, we have explored the impact of hand washing on the probability of infection when both the half-life of viral persistence as well as the hand contamination rate is varied in our supplementary material (page 32ff.). There we show that for long time intervals between hand washing events or long delays between hand washing and hand contamination events, hand washing is only effective with relatively low hand contamination events. This emphasises that reducing the hand contamination rate may be an effective alternative intervention (see page 34 in supplementary material).

Overall, I see this a very useful paper and my comments are aimed at allowing the authors to either refine their analysis or nuance their discussion, rather than seeing them as undermining their main conclusions.